

# Bottom topography effects on abyssal diapycnal mixing in the Eastern Mediterranean Sea

**Florian Kokoszka[1,2], Stefania Sparnocchia[2], Davide Cavaliere[3], Vincenzo Artale[3], Mireno**
**Borghini[4], Beatrice Giambenedetti[5], Federico Falcini[3*]**
[1]Stazione Zoologica Anton Dohrn, Villa Comunale, Naples 80121, Italy
[2]Istituto di Scienze Marine, Consiglio Nazionale delle Ricerche, Trieste 34149, Italy
[3]Istituto di Scienze Marine, Consiglio Nazionale delle Ricerche, Roma 00133, Italy
[4]Istituto di Scienze Marine, Consiglio Nazionale delle Ricerche, Lerici 19032, Italy
[5]Istituto Nazionale di Geofisica e Vulcanologia, Roma 00143, Italy
**\* Correspondence:**
Federico Falcini
federico.falcini@cnr.it
**Keywords:** Ionian Sea, diapycnal mixing, internal waves, shear and strain ratio, turbulent kinetic
energy.

## Abstract

The abyssal Ionian Sea is a deep region of interest for the entire ocean circulation of the Mediterranean
Sea, since it plays an important role in the ventilation processes of the whole basin. Here we investigate
spatial patterns of diapycnal mixing due to internal waves, over the bottom of the Ionian sub-basin. To
identify regional features of the internal wave field in terms of vertical shear and strain, we analyze
LADCP and CTD profiles, measured across the basin in 2007, covering various seafloor morphologies
(shelf, shelf break, and abyssal plain). Our results show that increasing seafloor roughness reduces the
variability of the shear-to-strain ratio, a pattern also influenced by correlations between slope and
roughness. Roughness appears to constrain waves toward higher frequencies, with high shear-to-strain
ratios associated with lower frequencies and flatter propagation angles, and low ratios linked to higher
frequencies and steeper beams. Spectral analyses indicate that rougher regions enhance strain variance
at small vertical scales while reducing shear variance at larger scales, leading to flatter shear spectra in
the low-wavenumber band. Together, these findings suggest that roughness redistributes energy from
large-scale toward small-scale, fundamentally altering the balance of internal wave energy across
scales. These results expand our insights for 3D ocean circulation models, providing useful knowledge
for ad hoc parameterization of mixing that should capture abyssal, internal wave–driven processes in
the Mediterranean Sea.





## 1    Introduction

The abyssal circulation, when particularly constrained by seabed topography, can bring to the generation of internal waves (IWs) that enhance finescale shear (i.e., vertical variation of the horizontal velocity) and strain (i.e., vertical gradient of isopycnal displacements) over rough bathymetry (Polzin et al., 1997). The resulting small-scale mixing, due to breaking of IWs, may cause upwelling of abyssal waters along sloping boundaries (Osborne and Burch, 1980; Walin, 1982; Wunsch and Ferrari, 2004; Garrett and Kunze, 2007; Nikurashin and Ferrari, 2013; Cavaliere et al., 2021; La Forgia et al., 2021). At basin scale, this implies that water parcels become heavier within the stratified ocean interior, while they result to become lighter along the oceanic boundaries; consequently, IW processes are likely responsible for transporting energy upward (Munk, 1966; Ferrari, 2014; Ferrari et al., 2016).

Despite its thorough implications in the ocean circulation, the relationship between the intensity of overturning circulation and deep, IW-induced mixing rates is not yet fully understood (Garrett and Laurent, 2002; Ferrari, 2014). Paucity of deep ocean measurements hampers our knowledge on abyssal mixing processes and, consequently, ocean heat content variability (Munk and Wunsch, 1998; Ferrari and Wunsch, 2009; Nikurashin and Ferrari, 2011; Waterhouse et al., 2014; de Lavergne et al., 2016; MacKinnon et al., 2017; Ferron et al., 2017; Artale et al., 2018). Few deep-ocean observations demonstrated that dissipation rates of turbulent kinetic energy, and the related changes of potential energy of the water column through diapycnal mixing, increase as the turbulent probes approached the ocean bottom above rough topography (Polzin et al., 1997; Ledwell et al., 2000; St. Laurent et al., 2012; Ferrari et al., 2016; Mashayek et al., 2017). The analysis of IW-field characteristics from microstructure and fine-structure observations suggests that bottom-generated IWs play a major role in determining the spatial distribution of turbulent dissipation (Sheen et al., 2013; Kunze et al., 2006; Ferron et al., 2014; Ferron et al., 2016). Recent observations indicate that IW field characteristics, shaped by local topography and forcing, show significant variability not only in amplitude but also in frequency composition, affecting the structure and energy distribution of the wave packets and the resulting turbulent mixing (Chinn et al., 2016). These considerations inspired us to investigate the potential role of IWs in mixing processes within the eastern part of the Mediterranean basin, where synchronous Lowered Acoustic Doppler Current Profiler (LADCP) and Conductivity, Temperature, Depth (CTD) data were available.

The Mediterranean Sea, due to its specific thermohaline, geographical, and morpho-bathymetric characteristics, as well as its intense convective and mixing processes, is one of the most interesting marginal seas of the global ocean (Schroeder et al., 2016; Artale et al., 2018). It is often considered as a "miniature ocean" due to its own thermohaline cell, where cold and/or salty waters sink during winter at specific regions and, subsequently, spread to intermediate and deep, near-bottom layers (Wüst, 1961; Bergamasco and Malanotte-Rizzoli, 2010; Millot and Taupier-Letage, 2005; Somot et al., 2006; Tsimplis et al., 2006). Moreover, among the Mediterranean-type climates, i.e., those defined by temperate, wet winters and hot/warm, dry summers over the western edges of five continents, the Mediterranean is the only one in which the atmospheric variability produces an intense deep-water formation (Seager et al., 2019).

Several analyses of the Mediterranean overturning circulation showed that this marginal basin is well ventilated, in comparison to the world ocean (Wüst, 1961; Malanotte-Rizzoli and Hecht, 1988;



Pinardi and Masetti, 2000; Theocharis et al., 2002; Artale et al., 2006; Malanotte-Rizzoli et al., 2014;
Schneider et al., 2014). Negative hydrological balance and cold-dry winds, blowing during the winter
season, make the Mediterranean Sea an extraordinary place of intense ocean water mass transformation
(Pinardi and Masetti, 2000), and thus, an ocean laboratory to study the main processes involved in the
global ocean circulation.
Surface water from the Atlantic Ocean enters the Mediterranean basin through the Strait of
Gibraltar (Figure 1a), forming a shallow overturning circulation at the upper layer. Flowing eastward,
along the African coasts, this Atlantic water increases its salinity and density, then sinking in the
Levantine basin in the eastern Mediterranean and forming the Levantine Intermediate Water (LIW).
The LIW flows back into the opposite direction, mixing with the surrounding water mass found along
its travel back and leaving the Mediterranean Sea through the Strait of Gibraltar, which works as a salt-
valve (Artale et al., 2006). Deep water exchanges between the western and the eastern Mediterranean
basins occur within the shallow Sicily Channel (Figure 1a), which makes the deep-water circulation of
the two sub-basins rather independent from each other (Sparnocchia et al., 1999; Napolitano et al.,
2003; Béranger et al., 2004; Schroeder et al., 2006). In the eastern basin, the Eastern Mediterranean
Deep Water (EMDW) is formed, alternately, in separate regions of the Eastern Mediterranean Sea
(Figure 1a), i.e., the southern Adriatic Sea and the Aegean Sea (Gačić et al., 2010). This cold and high
salinity winter water, after deep convection, flows into the Ionian Sea (Wüst, 1961; Roether and
Schlitzer, 1991; Schlitzer et al., 1991; Roether et al., 1996; Bensi et al., 2013a; Bensi et al., 2013b;
Bellacicco et al., 2016), i.e., the deepest sub-basin of the Mediterranean Sea, which has a maximum
depth close to the mean depth of the global ocean (Figure 1a). Such a thermohaline circulation is,
therefore, characterized by a superposition of intermediate and deep, zonal and meridional overturning
cells (Bergamasco and Malanotte-Rizzoli, 2010).
In the Ionian Sea boundary currents show intense downwelling due to their interaction with the
topographic constraints while the deepest part of the basin is characterized by extremely weak
stratification (Waldman et al., 2018), which induces a deepening and barotropization of the flow (Holte
and Straneo, 2017; Send and Testor, 2017). These deep waters are characterized by IW-turbulence and
mixing (van Haren and Gostiaux, 2011). Moreover, the Ionian Sea shows a semi-diurnal lunar tidal
(M2) and a dominant motion that is around the inertial frequency, mainly as freely propagating IWs
(Gerkema and Zimmerman, 2008). For this reason, the Ionian Sea constitutes a unique example for
studying the fundamental dynamics and processes of the abyssal layers in the Mediterranean basin,
also in the light of turbulent mixing due to IW breaking (Rubino et al., 2012; Artale et al., 2018;
Giambenedetti et al., 2024).
An intense observational activity was conducted in the Ionian Sea in the framework of the
Cubic Kilometre Neutrino Telescope (KM3NeT) project, from April 2006 to May 2009 (Kats et al.,
2006). During the KM3Net cruise (July 2007), hydrographic measurements were spread over a large
area, including a section that almost synoptically spanned the central Ionian Sea, from Sicily to Greece.
On the western side and up to the central abyssal plane, a marked stratification was found in the deep
layers, with a warmer and saltier water on the bottom, below a fresher and slightly colder layer (located
approximately at about 2750 m), probably due to deep water formation processes at different times
(Sparnocchia et al., 2011). Deep current meter measurements collected in the westernmost part in the
period 2007–2008 showed a quite energetic and impulsive abyssal circulation pattern, with maximum
velocities up to 15 cm/s, alternating with periods of quiet, strongly affected by the bathymetric





constraints. Furthermore, cyclonic and anti-cyclonic mesoscale structures, with a period of 5-to-11
days, superimposed on the background flow, were identified (Rubino et al., 2012; Meccia et al., 2015).

Artale et al. (2018), by analyzing water column characteristics of the Ionian Sea, observed
during the last three decades, focused on the hydrological processes occurring at the bottom layer. They
found that the ocean circulation in this abyssal plain is strongly affected by an interplay between
advection and diffusion. The progressive warming and salinification of this sub-basin produced warmer
near-bottom waters, causing an anomalous heat storage of ~1.6 W/m$^2$, i.e., a value three times larger
than the equivalent climate trend occurring in the same period at global scale (Bindoff et al., 2007).
The analysis of Artale et al. (2018) investigated the triggering of a diapycnal mixing due to rough
bathymetry. In particular, to explore topographic-induced mixing, these authors estimated dissipation
rates from the "CTD strain-based" parameterization (Kunze et al., 2006), showing the role of the sea
bottom in enhancing isopycnal vertical strain.

From the synergic use of LADCP and CTD data, collected in the Ionian Sea during the KM3Net
cruise in 2007, here we expand the analysis of Artale et al. (2018) and we seek to provide new insights
on the relation between IW mixing and bottom topographic constraints. In particular, we explore
different seafloor characteristics, in terms of bathymetric gradient and bottom roughness. LADCP
profiles, synchronous with CTD casts, allowed estimation of regional values of shear-to-strain ratio
($R_\omega$) within the parameterization of kinetic energy dissipation rate $\varepsilon_{iw}$ and, in turn, diapycnal diffusivity
$K_{iw}$ (Garrett and Munk, 1975; Munk, 1981; Kunze et al., 2006). In the frame of the Garrett-Munk
parametrization (Polzin & Lvov, 2011; See Data and methods), we explore the relation between $R_\omega$
and topographic features of the Ionian Sea. This approach employs CTD profiles to determine the
isopycnal vertical strain ($< \xi_z^2 >_{in\ situ}$), and LADCP profiles for the vertical shear of horizontal
velocities ($< V_z^2 >_{in\ situ}$), assuming that the variance is due to the presence of IWs (See Data and
methods). These variances are used to estimate the local energy's level of the IW field, to modulate a
canonical value of the dissipation rate of turbulent kinetic energy $\varepsilon_0$ (Garrett and Munk, 1975).

## 2    Data and methods
### 2.1    Hydrological casts

Our analysis is based on CTD and LADCP profiles (Borghini et al., 2025) acquired in July 2007
during a research cruise aboard the R/V Urania, conducted within the framework of the KM3NeT
project (https://www.km3net.org/). Hydrological stations were distributed over a wide area of the
northwestern Ionian Sea (Figure 1a), including strategic sites for the KM3NeT (Neutrino Telescope)
infrastructure that have been identified by the particle physics community (Katz, 2006).

CTD data were acquired from the sea surface to the bottom by using a SBE911-plus calibrated
before the cruise at the NATO center in La Spezia and were processed by the SBE Sea Soft program,
following the standard procedure suggested by the manufacturer. LADCP profiles were collected in a
subset of stations using a dual-head system installed on the rosette made of two RDI BB/WH 300 kHz
instruments looking upward and downward respectively. The instruments have been preset to acquire
20 bins with size 10.00 m and the LADCP data were processed using the LDEO LADCP software
Version IX.13 (https://github.com/athurnherr/LDEO_IX/releases/tag/IX_13), using CTD and
respective GPS fixes as auxiliary data.







## 2.2 Finescale parameterization

The small spatial scales (order of centimeters) and intermittent temporal nature (from minutes to hours)
involved in internal-wave-driven turbulent mixing, imply that direct measurements are difficult to
achieve. Indeed, measurements resolving small-scale turbulence rely on ship-based microstructure
profilers, not yet in wide use and, in any case, not on board during the 2007 KM3NeT research cruise.
To fill this gap, finescale parameterizations are being used to estimate the turbulent kinetic energy
dissipation rate and diapycnal diffusivity from more common instruments, such as CTDs, ADCPs, and
Argo profiles (Whalen et al., 2020; Polzin et al., 2014). Finescale parameterizations aim to infer the
centimeter-scale turbulent energy dissipation rate, operating on the intermediate [O(10-100) m] vertical
wavelengths that are assumed to mediate energy transfer between large and small scales in the ocean
(Polzin et al., 2014). Finescale parameterizations are formulated by reference to the Garrett-Munk 75
model (hereafter, GM) (Garrett & Munk, 1975; Cairns & Williams, 1976; Polzin & Lvov, 2011), which
provides an empirical expression for the internal wave energy density in the spectral domain. Strain-
and shear-based parameterizations allow to estimate the in-situ internal wave energy level, which
serves to establish a deviation from the GM model, and to adjust its canonical value for the turbulent
dissipation rate $\varepsilon_0$ to a more realistic estimate. In particular, the turbulent dissipation rate can be
expressed through a recent finescale parameterization proposed by Kunze et al. (2006) as:

$$\epsilon_{\mathrm{iw}} = \epsilon_0 \frac{N^2}{N_0^2} \left(\frac{E}{E_{GM}}\right)^2 F(R_\omega)L(f,N) \tag{1}$$


where $E$ is the in situ internal wavefield variance and $N$ the in-situ Brünt-Väisälä frequency, while
$E_{\mathrm{GM}}$, $\varepsilon_0 = 7 \times 10^{-10} W/kg$ and $N_0 = 5.2 \times 10^{-3} rad/s$ are the GM internal wavefield variance,
dissipation rate and Brünt-Väisälä frequency, respectively (Garrett and Munk, 1975; Munk, 1981).
Depending on whether the parameterization is based on shear or strain, $(E_{in-situ}/E_{GM})^2$ in Eq. (1) can
be estimated from the shear variance as $< V_z^2 >_{in-situ}^2 / < V_z^2 >_{GM}^2$, or from the strain variance as
$< \xi_z^2 >_{in-situ}^2 / < \xi_z^2 >_{GM}^2$.
The factor $L(f,N)$ in (1) represents the latitude effect, calculated as $L(f,N) = f \, cosh^{-1}(N/f)$ /
$f_0 \, cosh^{-1}(N_0/f_0)$ with $f$ the Coriolis parameter and $f_0$ the Coriolis's frequency at 30°. The term $R_\omega$
represents the internal wavefield's aspect ratio and the bulk frequency content, and it is defined as the
ratio of the buoyancy normalized shear variance to the strain variance:

$$R_\omega = (< V_z^2 >/ N^2) / < \xi_z^2 >), \tag{2}$$






whose effect on dissipation rates $\epsilon_{iw}$ is modeled by the term $F(R_\omega)$:

$F(R_\omega) = h_1(R_\omega) = 3/4(1 + 1/R_\omega)\sqrt{2/(R_\omega - 1)}$        (for shear-based parameterization)   (3a)
$F(R_\omega) = h_2(R_\omega) = R_\omega(R_\omega + 1)/(6\sqrt{2R_\omega - 1})$        (for strain-based parameterization)   (3b)

In Equations (3a,b), for $R_\omega = 3$ (i.e., the GM value), the correction functions are equal to 1 and do
not affect $\varepsilon_{iw}$. When $R_\omega > 3$, shear variance dominates, low-frequency internal waves contribute more
strongly, and $h_1$ decreases (underestimates $\varepsilon_{iw}$), while $h_2$ increases (overestimates $\varepsilon_{iw}$). When $R_\omega <$
3, strain variance dominates, high frequency internal waves contribute more to the ratio: $h_1$ increases
(overestimates $\varepsilon_{iw}$), $h_2$ decreases (underestimates $\varepsilon_{iw}$). This makes the shear-based parameterizations
more reliable when $R_\omega > 3$, and conversely the strain-based parameterizations are more reliable when
$R_\omega < 3$ (Chinn et al. 2016).
The determination of $R_\omega$ requires, ideally, shear and strain variance measurements through LADCP
and CTD observations, respectively. The strain $\xi_z$ is defined as the vertical derivative of the isopycnal
displacement and is generally calculated as $\xi_z = (N^2 - N^2{}_{fit})/\overline{N^2}$, where $N^2$ is the Brünt-Väisälä
frequency, $\overline{N^2}$ a mean value over the vertical segment of water column, and $N^2_{fit}$ a polynomial quadratic
fit: $N^2$ - $N^2_{fit}$ represents the IW's density perturbation, normalized by $\overline{N^2}$ to obtain the vertical
derivative of the isopycnal displacements.
Instead of determining $\xi_z$, we follow the approach of Ferron et al. (2014) and we determine $\xi$ from
the density fluctuations as $\xi = (g/\rho_0)(\rho_{HP}/\overline{N^2})$, with $g/\rho_0$ the standard gravity on the reference
density, and $\rho_{HP}$ density filtered by a high-pass Butterworth filter with a cutoff wavelength of 160m to
remove the density fluctuations associated with the background stratification, whose signal would
contaminate the integration of the strain spectra (Kunze et., al 2006). This advantageously avoids fitting
a polynomial to $N^2$ (e.g., quadratic fit in Kunze et al., 2006), that could be inconsistent with the shape
of the density profile in case of strong vertical gradients. Strain variance is then calculated as the
integral of the isopycnal displacement spectra between $k_{min}$ and $k_{max}$:

$< \xi_z^2 >= \int_{k_{min}}^{k_{max}} k^2 S(\xi) dk.$                                                                       (4)

Similarly, the shear $< V_z^2 >$ is obtained from the velocity spectra as

$< V_z^2 >= \int_{k_{min}}^{k_{max}} k^2 S(u, v) dk.$                                                                   (5)




To integrate the spectra, $k_{min}$ and $k_{max}$ must be carefully determined, as shear and strain rely on
different instrument limitations and can be sensitive to distinct phenomenological contributions.
In general $k_{min}$ is fixed by the choice of the data segment length $k_{min} = 2\pi/L \ rad \ m^{-1}$ , where $L$ is
generally of 200-to-500 m, i.e., a compromise between the larger vertical wavelengths to be considered
(the spectral resolution to be employed) and the desired vertical resolution. For strain, in particular,
the integration is performed from $k_{min} = 2\pi/O(150m)$ (Kunze, 2006, Pollman, 2017; 2020), to
exclude contributions from larger-scale background stratification, and continues up to $k_{max} =$
$2\pi/10m$ (the GM upper limit).
For shear, a velocity noise is expected to dominate at high wavenumber, especially in weakly
stratified layers lacking turbulent microstructures or suspended material ("water-column reflectors")
(Kunze et al., 2006), which tend to reflect the acoustic signal emitted by Doppler profilers like the
LADCP. Their absence reduces the quality of the acoustic return, leading to higher measurement
uncertainty and potentially spurious velocity variance, especially at small vertical scales. Thus,
wavenumbers larger than $k_{max} = 2\pi/O(100 - 50m)$ represent the upper limit range of usable
vertical wavenumbers for integration.
An additional saturation criterion is applied to limit the integration of spectra (Gargett, 1990, Kunze
et al., 2006). Specifically, the integration is stopped when reaching the saturation value of $0.66 \ N^2$ for
the shear variance and $0.22 \ N^2$ for strain. This threshold represents the onset of internal wave breaking
or turbulence, where the linear internal wave regime breaks down. The corresponding vertical
wavenumber $k_c$ at which this value is reached is retained as the upper limit for the integration. This
approach constrains the finescale parameterization to physically realistic shear and strain levels,
avoiding overestimation of internal wave energy due to instrument noise or unresolved scales,
particularly in regions of enhanced energy.
In our case, strain and shear are derived from CTD and LADCP measurements from the bottommost
640 m of each vertical profile. Signals are then linearly detrended and Fourier transformed with a
Hamming tapering window. Strain and shear variance estimates are obtained by integrating the 640 m
segments, overlapping every 10 m from the bottom to the surface, to account for non-homogeneity in
the statistical distribution of the water column properties and to have a more robust representation of
the fine-scale variability. Conveniently, final variances are obtained by averaging collections of
estimates over finite-sized segments (here 80 m).
The shear integration upper limit has been identified from the velocity error. The LADCP processing
(velocity-inversion method (Polzin, 2002), LDEO), provides a single-bin error profile for each station.
The average $\overline{V_{ERR}} = 0.075 \ m.\ s^{-1}$ is used to estimate a velocity error by segment as $V_{NOISE} =$
$\overline{V_{ERR}}/\sqrt{640m/10m}$. Its associated variance $V_{NOISE}^2$ is then distributed on the spectral domain to
produce a white noise velocity spectrum that, once sheared, allows to obtain a shear noise spectrum
that will identify the noise-free upper limit in the spectral domain.
The resulting spectra are then averaged by stratification bins, as shown in Figure 1b, where we
highlighted their overall shape and the noise-free bandwidths we retain for integration, from 640 to
107 m. This choice is conservative, as our lowest stratification levels are close to the $N_{ERR} = 5 \cdot 10^{-4}$
s[-1] (discussed in Kunze, 2006), and consistent with the integration set up of Ferron (2014) that stopped
the integration at 107 m in the most limitative case. Additionally, this treatment combined both data





from downward and upward profiles, in relatively small bin size (10 m), limiting then the expected
variance loss at short scales due to the various steps of data processing (Polzin, 2002; Thurnherr, 2012).
For strain, integration is led from 128 to 10 m (the GM upper limit).
Since shear and strain are integrated on their respective bandwidths, instead of using Eq. (2) directly,
the shear-to-strain ratio is calculated as:

$$R_\omega = R_\omega{}^{GM}(<V_z^2>/<V_z^2>_{GM})/(<\xi_z^2>/<\xi_z^2>_{GM}), \qquad (6)$$

where $R_\omega{}^{GM} = 3$, both shear and strain factors are divided by their associated value in the GM model,
evaluated on the same spectral bandwidth, respectively. Finally, from Equations (1), (3), and (6), the
kinetic turbulent diffusion rate $K_{iw} = \Gamma\varepsilon_{iw}/N^2$ is calculated with the Osborn-Cox relation (Osborn
and Cox, 1972), using a mixing efficiency value of $\Gamma = 0.2$.
All functions used for obtaining Equation (6) are in Kokoszka, F. (2025).

**2.3    Internal wave beams and impact of bathymetry on $R_\omega$**
Traditional finescale parameterizations of turbulent kinetic energy dissipation rely on the
assumption that the internal wave field follows a GM-like structure. In this framework, a fixed value
of the shear-to-strain ratio, typically $R_\omega \approx 3$, is used to represent the relative contributions of vertical
shear and isopycnal strain to the finescale energy budget. This fixed value implicitly assumes a
broadband, statistically stationary internal wave field, with no significant spatial variability in spectral
composition. However, these models sometimes fail to capture the true dynamics near mixing hotspots
where the internal wave spectra can be significantly distorted, showing bias toward higher or lower
frequencies.
Accumulating evidence from both observations and modeling (Ijichi and Hibiya, 2015, 2017;
Takahashi, et al. 2021; Dematteis et al., 2024) indicates that this assumption is often violated, especially
in regions influenced by complex bathymetry, variable stratification, or energetic boundary currents.
In these settings, the internal wave spectrum is distorted, leading to departures from the canonical
balance between shear and strain. As a result, $R_\omega$ can vary significantly in space and time, reflecting
changes in the dominant frequency content of the wave field.
Ijichi and Hibiya (2017) used 3D eikonal equations to simulate how internal waves transfer energy
across distorted spectra, including variations in the wave energy level, the local buoyancy frequency,
and the inertial frequency. Their results confirm that energy transfer rates are consistent with the
Henyey et al. (1986) model, which predicts dissipation from internal wave-wave interactions and
highlights the importance of the spectral shear-to-strain ratio $R_\omega$. The study questions the accuracy of
existing finescale parameterization models, particularly in environments with distorted spectra near
boundaries or topographic features. To address these limitations, Ijichi and Hibiya (2015, 2017)
propose a revised parameterization that explicitly considers both broadband and narrowband spectra.





This leads to more accurate dissipation estimates by incorporating a dynamically varying $R_\omega$ instead
of assuming a constant value based on idealized spectral shapes. Their approach was further developed
by Takahashi et al. (2021), who showed how vertical wavenumber spectra distortions (such as spectral
humps) can bias traditional estimates, emphasizing the need to resolve the spatial variability of $R_\omega$.
From a physical point of view, $R_\omega$ represents the ratio between the buoyancy-normalized vertical
shear variance and the isopycnal strain variance. The spectral structure of the internal wave field
determines whether energy is primarily stored in the shear (associated with horizontal motion) or in
the strain (associated with vertical isopycnal displacement). The theoretical foundation for this lies in
the internal wave dispersion relation:

$$\omega^2 = \frac{N^2 k_h^2 + f^2 k_z^2}{k_h^2 + k_z^2},$$   (7)

where $\omega$ is the wave frequency, $\boldsymbol{k}_h = (k_x, k_y)$ and $k_z$ the horizontal and vertical wavenumbers,
respectively.
Equation (7) shows the intrinsic relation between horizontal (vertical) wavenumber and the shear-to
strain ration. When $\omega \approx f$, it follows that $k_z \gg k_h$ and that the wavefield is dominated by horizontal
motion, yielding high shear $\partial u \,/\, \partial z$, i.e., the wavefield varies rapidly with depth and more slowly in
the horizontal direction. This configuration results in particle motions that are predominantly horizontal
but vary significantly over short vertical distances. Consequently, the vertical gradient of horizontal
velocity (i.e., the vertical shear) is amplified. Even when vertical displacements of isopycnals are small,
the horizontal velocity field can exhibit strong vertical variation due to the short vertical wavelength
(large $k_z$). This sharp vertical structure leads to high shear variance.
In realistic internal wave spectra, low-frequency components (near the Coriolis frequency $f$) tend to
concentrate energy in horizontal motions that change rapidly with depth, making shear-dominated
fields a hallmark of low-frequency wave regimes.
Conversely, when $\omega \approx N$, $k_h \gg k_z$ and the particle motion becomes primarily vertical, enhancing
isopycnal displacement and thus the strain $\partial \xi / \, \partial z$ (i.e., significant vertical displacements of
isopycnals). It results that the wave propagates with a nearly vertical beam and the wave field varies
slowly with depth. For this condition, although the vertical structure of the wave is broad (i.e., fewer
oscillations along z), the amplitude of vertical displacements is large. As a result, the isopycnals
experience stronger stretching and compression, leading to enhanced strain energy. This explains why
internal waves at high frequencies, despite their low vertical wavenumber, contribute
disproportionately to the strain variance. The enhanced vertical motion associated with high-frequency
waves increases the spatial variability of $\xi$, and thus the observed strain, even if $\partial \xi / \partial z$ is not large in
the sinusoidal sense. In real oceanic spectra, high-frequency components carry more energy in vertical
displacement, making the strain-dominated regime a hallmark of high-frequency wavefields.
A propagation angle of an internal wave ray can be determined using the dispersion relation:




$$\frac{k_z{}^2}{k_x{}^2+k_y{}^2} = \frac{N^2-\omega^2}{\omega^2-f^2};$$ (8)


thus,

$$tan^2\alpha = \frac{N^2-\omega^2}{\omega^2-f^2}.$$ (9)


From Equation (9), the angle of the wave trajectory (i.e., the beam angle $\alpha$) relative to the horizontal
can be determined. The angle $\alpha$ describes a cone of propagation that supports the wave:

$$\alpha = tan^{-1}\sqrt{\frac{N^2-\omega^2}{\omega^2-f^2}}.$$ (10)


The wave energy propagates with the group velocity $\boldsymbol{c}_g$, perpendicular to the wave direction $\boldsymbol{k}$, at
an angle $\beta = 90° - \alpha$. For $R_\omega$ it is possible to derive the following expression (see Kunze et al,.
2006; Ijichi and Hibiya, 2015):

$$R_\omega = \frac{\langle V_z^2\rangle}{\langle \xi_z^2\rangle N^2} = \frac{\omega^2+f^2}{\omega^2-f^2} \simeq 1 + \frac{2f^2 k_z^2}{N^2 k_h^2},$$ (11)


from which we estimate a local value of $\omega$ in the water-column, given $N(z)$ and the local $f$, that we
assume representative of a certain regional extent on the horizontal. From $\omega$ we can estimate $\beta$
perpendicular to $\alpha$. The relation between $R_\omega$, $\omega$ and $\beta$ will be exploited later in our study to associate
the beam angle with $R_\omega$ value and compare it with the bathymetrical slope to infer propagation insights.
Equation (10), along with the physical interpretation of Equation (7), highlights that the wave geometry
(i.e., the ratio $k_z^2/k_h^2$ ) dominates the shear-strain dynamics: for $k_z \gg k_h$ we have high shear and $R_\omega$
values, and for $k_h \gg k_z$ high strain values and low $R_\omega$. In other words, high $R_\omega$ values correspond to
low-frequency, shear-dominated conditions, while low $R_\omega$ indicates high-frequency, strain-dominated
fields.
A growing body of evidence shows that turbulent mixing is significantly enhanced in regions of
rough bottom topography and sloping boundaries (Polzin et al., 1997; Ledwell et al., 2000; Kunze et
al., 2006; Ferrari et al., 2016; Mashayek et al., 2017). Internal wave interactions with these features



can lead to reflection, scattering, and energy transfer toward higher vertical wavenumbers, increasing
the likelihood of wave breaking and dissipation (Nash et al., 2004; Legg & Adcroft, 2003; Musgrave
et al., 2022). In addition to tidal processes, geostrophic flows impinging on small-scale topography
generate broadband internal waves that can radiate and dissipate energy locally, contributing to abyssal
mixing (Nikurashin & Ferrari, 2010a, 2010b). These mechanisms, collectively, explain much of the
spatial variability in observed mixing rates and highlight the importance of accurately representing
topographic interactions in ocean mixing parameterizations.
When an internal wave encounters a rough topography, its propagation is disturbed, promoting
scattering, reflection, or the generation of higher-frequency waves. This tends to break down low-
frequency, large-scale wave structures and shift spectral energy toward higher vertical wavenumbers.
Since strain is associated with large vertical displacements, characteristic of high-frequency waves,
this shift leads to a systematic decrease of the shear-to-strain variance ratio $R_\omega$ in rougher regions, as
documented in Kunze et al. (2006), where departures from the canonical Garrett–Munk spectrum were
observed in high-roughness environments. As a result, areas of high roughness typically exhibit lower
$R_\omega$, indicating increased strain. In contrast, smoother topographic regions preserve the lower-frequency
shear-dominated character, yielding higher $R_\omega$.
In our investigation of the impact of bathymetry on $R_\omega$, we analyze the KM3NeT hydrological
stations in relation to bathymetric features such as slope and topographic roughness, calculated using
the GEBCO gridded data (https://www.gebco.net) over the Ionian Sea area, at 15 arc-second intervals.
In particular, slopes are estimated between the hydrological cast position and the points located apart
in the eight cardinal directions around, then the average is retained. A distance of 10 km between the
two points is chosen to avoid overestimation of the topographic slope, due to the presence of small
bathymetric features. Topographic roughness is calculated from the variance of the bathymetry over
the same neighboring area (i.e., 20-by-20 square kilometers).

**3    Results**
To investigate the influence of bottom topography on internal wave shear-to-strain ratio from
Equation (6), our dataset is bin-averaged every 80 m, from bottom to surface. From the first upward
segment of 640 m we obtain values at the center of the segment (i.e. 320 m from bottom), and the
associated first binned value is available into bin 4, centered at 360 m from bottom. We stop then at
bin 12 (centered 920 m from the bottom). By doing this, we consider 8 bins of 80 m (Figure 1b).
Results show that most of the near-bottom values of $R_\omega$ are larger than the canonical value of 3,
given by the GM model (Figures 1 and 2a; Supplementary Table S1). Smaller values, from 1.01 to 10,
are generally observed in the shelf break regions, as well as at some deep stations (Figures 1 and 2a;
Supplementary Table S1), or close to the Malta escarpment, where mesoscale bottom meandering and
vortical structures occur (Sparnocchia et a., 2011; Rubino et al., 2012). For deep, offshore casts,
estimations show large values. An increase of shear-to-strain ratio is evident in the western part of the
abyssal plain, between 2000 and 3000 m depth, i.e., along the offshore portion of the three cross-shore
transects T1, T2, and T3 (Figure 1a and 2a). Here, $R_\omega$ shows values between ~10 and ~30 (Figures 1
and 2a; Supplementary Table S1).





$R_\omega$ values are then compared with the local bathymetric roughness, i.e., the logarithm of the variance
of seafloor elevation (Figures 2a,b). In general, it results that regions with higher roughness show lower
$R_\omega$, suggesting, for those regions, a shift toward higher-frequency energy and enhanced vertical
displacement (i.e., strain). High values of bottom roughness are distributed along the eastern and
northern shelf; a smoother bathymetry is recognized in the abyssal plain. Correspondingly, $R_\omega$ displays
lower values (<10) in areas of high roughness, while offshore regions are associated with higher values
(>10). By plotting $R_\omega$ against bathymetric slope and roughness (Figure 2c), we find that although a
linear relationship is difficult to establish, an interesting pattern emerges: increased roughness appears
to reduce the dispersion of $R_\omega$. This effect could also be attributed to the bathymetric slope, as slope
and roughness are correlated, with steeper zones leading to greater variance in elevation. We
hypothesize that this pattern indicates waves being constrained toward higher frequency bands by the
roughness.
Indeed, further analyses show how the increasing $R_\omega$ corresponds to decreasing frequency and flatter
beam angles (i.e., more horizontal propagation), while low $R_\omega$ indicates steeper beams and higher
frequencies (Figure 3a). The vertical line and shaded area in Figure 3a highlight the mean and observed
range of $R_\omega$ in our dataset, contextualizing the frequencies and angles involved. This framework sets
the stage for comparing beam angles with local bathymetric slopes. A Δ-slope is inferred as the
difference between the bulk beam angle in Equation 10 and the bathymetry slope. This angle controls
the reflection regime: near-critical reflection (Δ-slope ≈ 0°) concentrates energy in horizontal shear,
while supercritical conditions (Δ-slope > 0°) promote upward-propagating waves and stronger vertical
displacements (strain) (Musgrave et al., 2022). Once scattered through this quantity, a clear pattern
appears, i.e., $R_\omega$ is clearly related to Δ-slope (Figure 3b). Higher $R_\omega$ values correspond to smaller angle
differences, while lower $R_\omega$ values are associated with steeper conditions. This indicates a shift in
frequency content, illustrated by the associated ω-distribution (Figure. 3c), with frequencies increasing
as Δ-slope grows. Additionally, roughness appears to constrain the dynamical range of $R_\omega$ dispersion.
This suggests a critical threshold where high frequencies and smaller scales begin to dominate over
shear.
By organizing the shear and strain spectra into bins of roughness and Δ-slope to identify whether
specific bandwidths are affected, one might expect enhanced signals at larger vertical scales (≈300 m)
or smaller scales (≈100 m), depending on the preferential frequency content potentially enhanced by
the bathymetric features. The resulting spectra (Figure 4a) show that strain variance increases at high
vertical wavenumbers (small vertical scales) in those regions characterized by elevated roughness,
while shear variance diminishes at low wavenumbers, reflecting a redistribution of energy across
scales. Indeed, shear spectra under rougher conditions (blue spectra in Figure 4a) appear flatter in the
lower wavenumber band compared to smoother regions. Intermediate roughness levels (green spectra
in Figure 4a) show increased energy around vertical scales of 64 m, potentially indicating enhanced
energy distribution at shorter scales. Strain spectra exhibit a distinct pattern for strong roughness, with
higher energy levels compared to other bins. These observations suggest that increasing roughness
reduces energy distribution at larger scales (shear lower band) while enhancing strain variance
contributions at smaller scales (upper band).
To isolate the influence of slope geometry, we use the beam-to-slope angle difference Δ-slope. The
shear spectra in Figure 4b show a clear dependence on Δ-slope. Indeed, higher $R_\omega$ values correspond
to smaller angle differences, while lower $R_\omega$ values are associated with steeper conditions. This
indicates a shift in frequency content, with frequencies increasing as Δ-slope grows. Spectra in Figure
4b support what we observed in Figure 3, suggesting a critical threshold where high frequencies and



smaller scales begin to dominate over shear. This interpretation is further confirmed by the patterns
observed in Figure 5: as Δ-slope approaches 0, lower frequencies become dominant, indicating more
horizontal motion. This is associated with increased shear and a higher prevalence of critical
Richardson instability events (Figure 5a,b). The deviation from the GM model is evident in the shear
ratio, with values exceeding 8–10 times the model (Figure 5c,d), while the strain consistently remains
below the GM model's predictions (Figure e,f), and vice-versa.
Regarding dissipation or diffusion rates of turbulent kinetic energy, we present their associated
scatterplot and geographical distributions in Figure 6. Spatial pattern of diapycnal diffusivity ($K_{iw}$;
Figure 6f) shows that our estimates are close to the typical "low" observed ocean values of $10^{-5}$ or
below ($10^{-6}$) (Munk, 1966; Kunze and Toole, 1997; Polzin et al., 1997), between 1000 and 2500 m
depth, on the northern part of the area (e.g., stations NK30, NK31 of transect T1 and stations NK26,
NK25, and NK24 of transect T2; Figures 1a; Supplementary Table S1).  Intensified area corresponds
to roughest zones with values spanning above 5 $10^{-5}$ up to closer to $10^{-4}$ (NK32, NK21, NK22, NK27,
NK13, 11 NK17, NK6; Figures 1a; Supplementary Table S1). In the abyssal layer, the function $F(R_\omega)$
in Eq. (3), in becoming smaller than 1 for large values of $R_\omega$, contributes to inhibit the terms of the
parameterization (see Data and methods). In this layer IW mixing is expected to be weak. Although
$\varepsilon_{iw}$ and $K_{iw}$ obviously depend on $R_\omega$, scatter plots in Fig 6a-d show how intense values of dissipation
or diffusion rates are coherently distributed along Δ-slope values.

**4      Discussions and conclusions**

Mixing in the ocean can be triggered by the combination of boundary constraints and
hydrographic conditions (Marotzke and Scott, 1999; Kuhlbrodt, 2008), and it is concentrated above
seamounts, mid-ocean ridges, and along/cross strong ocean currents (Polzin et al., 1997; Ledwell et al.,
2000). In these conditions, diapycnal mixing contributes to transfer and redistribute water masses and
heat throughout the deep ocean (Polzin et al., 1997; Ledwell et al., 2000; Marotzke and Scott, 1999;
Levitus et al., 2000; Artale et al., 2018). By increasing the potential energy within a stratified fluid,
and thus by raising the water mass center on a larger time and spatial scale, diapycnal mixing maintains
the long-term baroclinic balance of the water column (Wunsch, C., & Ferrari, 2004; Munk and Wunsch,
1998; Iudicone et al., 2003), thus contributing to the general ventilation of ocean basins. Investigations
on turbulent vertical mixing in the deep sea revealed that abyssal circulation and the related dynamical
effects of deep upwelling are more complex than what was envisioned during the twentieth century
(Stommel and Arons, 1959; Polzin et al., 1997).

Our analysis in the Ionian Sea contributes to the investigation of the IW mixing, as described
by Kunze et al. (2006). We focused on hydrological stations where we took advantage of synchronous
LADCP and CTD profiles from which we infer regional values of the near-bottom ratio between shear
and strain variances ($R_\omega$). Values for $R_\omega$ are necessary for deriving both dissipation rate of turbulent
kinetic energy and diapycnal diffusivity (i.e., $\varepsilon_{iw}$ and $K_{iw}$, respectively). Our goal, in particular, was
to quantitatively assess the spatial patterns of shear-to-strain ratio to highlight the role of morpho-
bathymetric features in enhancing dissipation and diffusion rates, in terms of depth, slope, and
roughness.
We found a general relation between bottom depth and shear-to-strain variance ratio ($R_\omega$),
which tends to increase from the shelf break to the abyss: large (small) value of $R_\omega$ is found over small
(large) topographic slope and/or low (high) roughness of the seafloor, consistent with the hypothesis
that gentle (steep) slopes and/or low (high) roughness would generate low (high) frequency internal





waves, leading to the increase in the shear (strain) contribution. In particular, for the northwest group
of stations (i.e., the shelf-to-plain transects T1, T2, and T3; Figure 1a), we found coherent trends of
$R_\omega$, which co-varies with depth, slope, and roughness. Among those stations, high values of $R_\omega$ tend
to inhibit both $\varepsilon_{iw}$ and $K_{iw}$.
Our observations bring to hypothesized that, in rougher conditions, the intensified energy
cascade and dissipation related to the increase in local shear, tend to break down coherent vertical
displacements associated with IWs. This can result in a suppression of the strain spectra in these
rougher areas. The imbalance observed (enhancement of shear variance and lower strain variance, with
respect of the trend observed for the other roughness cases) reflects a wave field that is more turbulent
and less coherent (Chinn et al, 2016).
Although in abyssal waters (i.e., deeper than 3000 m), specific stations of transect T4 (e.g., from
Station NK6 to Station 11; Figure 1a) show a behavior similar to the one observed in the shelf-to-plain
transects: $R_\omega$ increases with distance from the shelf break, starting from $R_\omega \sim 10$, although we are in
abyssal waters. We notice, however, that those abyssal stations of Transect 4 that show low values of
$R_\omega$ (i.e., values we expect in shallow areas) are close to the topographic constraints of the Malta
Escarpment (Figure 1a), where energetic mesoscale meandering features and vortical structures at the
bottom, were observed (Rubino et al., 2012; Meccia et al., 2015). These low values of shear-to-strain
ratio may be due to the eventual mesoscale eddy–internal wave coupling that would represent a sink
of eddy energy and then a source of IW energy (Polzin, 2010; Takahashi et al., 2021). These abyssal
stations in Transect 4 may therefore represent those particular, "nontraditional" cases (Gerkema and
Shrira, 2005) where deep mesoscale activity is able to trigger diapycnal mixing by breaking of IWs.
Kunze et al. (2006) suggested that the lack of reflectors in the water column at deep locations
could lead to an overestimation of the shear variance, reflected ultimately in strong values of $R_\omega$.
However, the in-situ spectra we analyzed do not indicate such a contamination.
In particular, we observe that the bottom layer of the eastern abyssal stations is prone to double-
diffusion processes, which may not make our analysis reliable: stations NK17, 12, and 13 (in transect
T4), station 25 (Transect 2), and stations 32 and 34 (Transect 1) are those that show more than 60% of
the near-bottom water column, with a Turner angle $-90° < Tu < -45°$ (i.e., diffusive convection regime)
and $45° < Tu < 90°$ (salt fingering regime), and a Richardson number $Ri < 0.25$ (Miles, 1961; Ruddick,
1983; Thorpe, 2005). An additional analysis of the density ratio $R\rho$ (not shown) indicates that only a
few thin layers fall within the "active" ranges typically used in double-diffusion parameterizations
(0.5–1 for diffusive convection and 1–3 for salt fingering). This suggests that, while double-diffusive
processes may occur locally, their influence on the overall analysis is likely limited.
Baroclinic tide and lee wave generation by tidal and geostrophic flows impinging on rough
topography are among the most significant sources of IW energy for the deep ocean (Egbert and Ray,
2000; Garrett and St. Laurent, 2002; Cavaliere et al., 2021; La Forgia et al., 2021). Our results reinforce
the concept of IW generation from the interaction between deep, geostrophic currents and the
topographic constraint and roughness, which in turn may cause local breaking processes. In assessing
an actual, systematic morpho-bathymetric characterization of the shear and strain variances, we also
highlight horizontal gradients of dissipation rate and diapycnal diffusivity we observe. This results in
horizontal gradients of vertical mixing that affect thermohaline gradients that, in turn, may trigger
additional components for the bottom ocean circulation (Jones and Abernathey, 2019).
Finally, our analysis aims to provide useful insights for ocean circulation models, which are
often too sensitive to vertical eddy diffusivity and are largely affected by inaccuracy at deep layers
(Gargett and Holloway, 1982; Wright and Stocker, 1992). In particular, the large variability of the
shear-to-strain ratio we found, may help the parametrization of mixing that should capture the



additional abyssal flow from the inclusion of wave–driven mixing (Simmons et al., 2004; Saenko and
Merryfield, 2005; Oka and Niwa, 2013; Melet et al., 2014; De Lavergne et al., 2016).

**5    Conflict of Interest**

The authors declare that the research was conducted in the absence of any commercial or financial
relationships that could be construed as a potential conflict of interest.

**6    Author Contributions**

FF, VA, and FK delineated, supervised the study. SS, MB, and FF collected the data. SS and FK
processed and analyzed the CTD and LADCP data. FK, SS, VA, DC, and FF contributed with
resources, analyzed the data and wrote the manuscript.

**7    Funding**

This work was partially supported by EMODnet (European Marine Observation and Data Network
Physics), the Copernicus Climate Change Service (C3S) project, and the Flagship Project RITMARE
(The Italian Research for the Sea), coordinated by the Italian National Research Council and funded
by the Italian Ministry of Education, University and Research.

**8    Acknowledgments**

Data were collected in the framework of the project ''Specific Support Action for the Design Study
for a Deep-Sea Facility in the Mediterranean for Neutrino Astronomy and Associated Sciences''
(KM3NeT), EU contract no. 011937. We thank Daniele Iudicone, Manuel Bensi, and Enrico
Zambianchi for the insightful discussion at the initial stage of the investigation.

**9    Data and Code Availability**

Data are available on Sea Scientific Open Data Publication (SEANOE),
https://www.seanoe.org/preview/108742?token=NkpdOxO5966RjZ--Rqdw1c0dCZ-6iGSW
Codes that were used for this analysis are available at
https://zenodo.org/records/17170422 (DOI: 10.5281/zenodo.17170421)



**Figures**

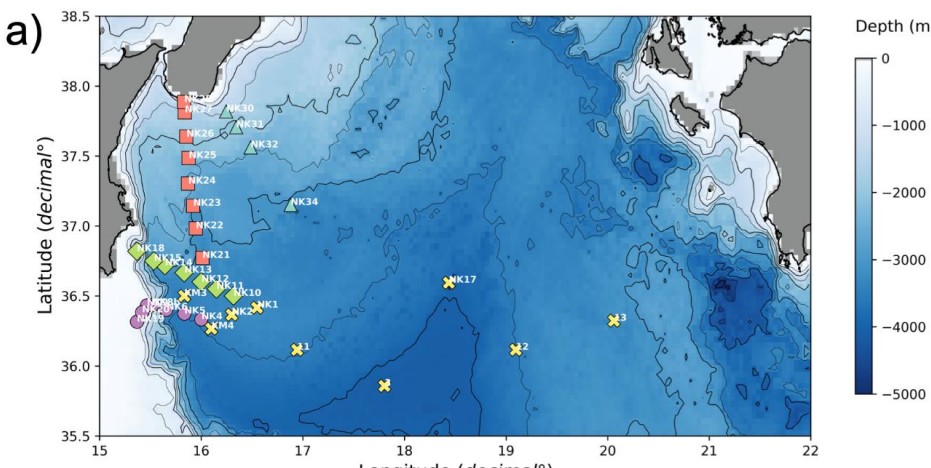

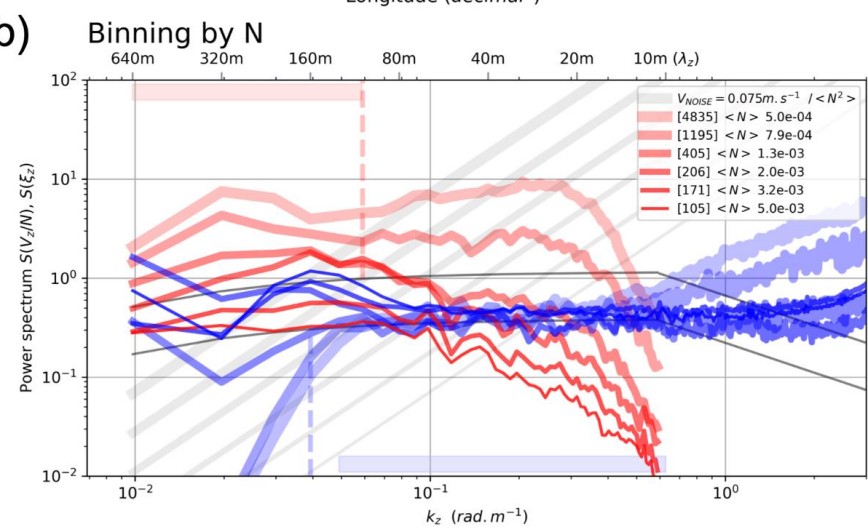

**Figure 1. a)** The Mediterranean Sea bathymetry and the main regional features that are discussed within the text; LADCP-CTD casts are grouped in four main hydrological transects: three shelf-to-plain transects, i.e., T1 (▲), T2 (■), and T3 (♦) and one abyssal transect T4(✗). **b)** Buoyancy-normalized shear (red) and strain (blue) wavenumber spectra, with their associated GM model (upper black line: shear; lower black line: strain). Spectra are averaged by bins of stratification (thick to thin, from weak to higher stratification). Number of spectra by bins, with the average N of the bin are reported in legend. Shear noise spectra associated with a noise velocity of 0.075 m/s are indicated in gray (thickness indicates its association to the N bins). Instrumental wavenumber limits are indicated with the vertical dotted lines: upper limit of $2\pi/128 \ rad \cdot m^{-1}$ for shear due to noise contamination; lower limit of $2\pi/160 \ rad \cdot m^{-1}$ for strain to filter the large-scale background stratification. Retained wavenumber bandwidths for integration are indicated with the shaded ranges on top and bottom (640-107 m for shear; 128-10 m for strain).



**Figure 2**. **a)** Shear-to-strain ratio $R_\omega$ at the bottom of the LADCP-CTD profiles, averaged from bins
of 80m in the layer 360-920m distant from the bottom; low (high) values of $R_\omega$ correspond to high-
(low-) frequency, strain-dominated (shear-dominated) regimes. **b)** Bottom roughness (log10 of var(z))
over the study area. **c)** Scatterplot of $R_\omega$ as a function of bathymetric slope (x-axis), and roughness (z-
axis, in color).

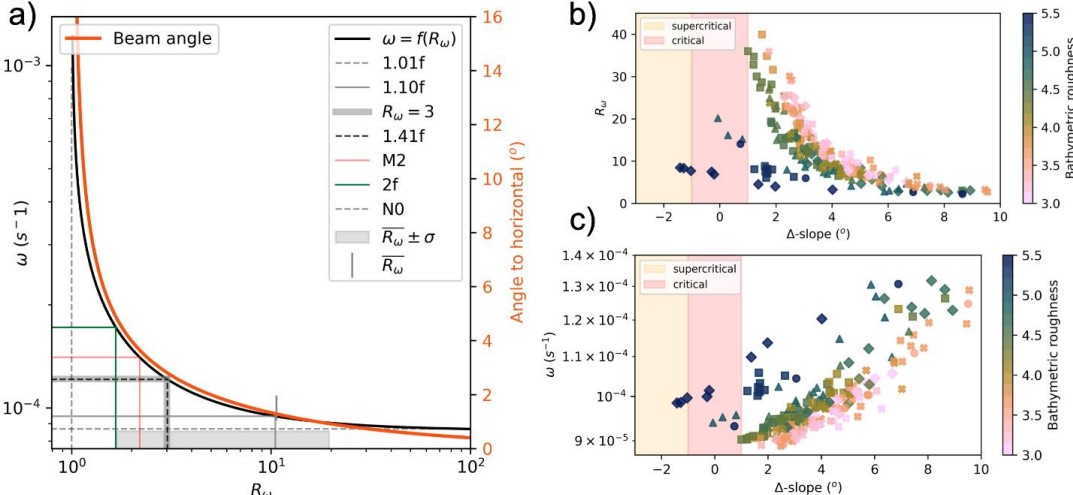


**Figure 3. a)** Correspondence between $R_\omega$ and ω (black), and the beam angle associated with $R_\omega$ (orange), computed at the average latitude of the study (36.75ºN). Typical values, from high to low $R_\omega$, are reported. The average $R_\omega$ value of our study is indicated with the vertical line marker, and its distribution range is indicated by the gray area. **b)** Distributions of $R_\omega$ and **c)** ω in the layer 360-920 m distant from bottom, in function of the difference between the associated beam angle and the bathymetry slope (Δ-slope). Roughness is indicated through the z-axis (in color). Higher values on the x-axis indicate a beam angle steeper, relative to the bathymetry slope, while lower values indicate a beam aligning with the slope. The critical range where the alignment is close to be parallel to the slope is arbitrarily marked by the shaded red area from -1 to 1 degrees, out of which a supercritical range is indicated in orange.




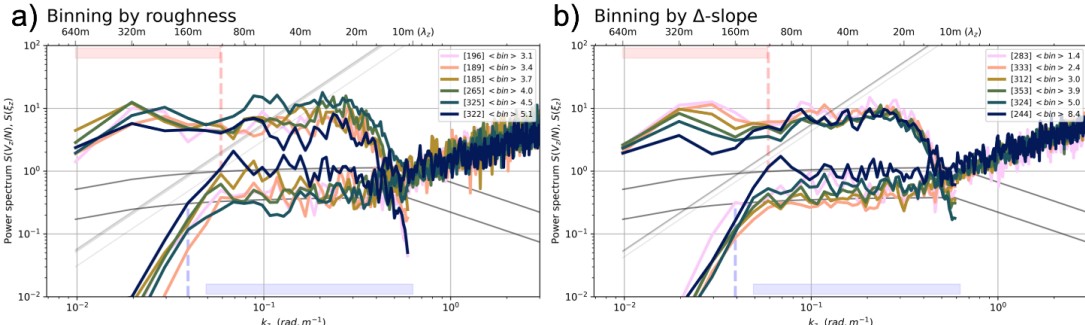

**Figure 4.** Buoyancy-normalized shear **(top)** and strain **(bottom)** wavenumber spectra, with their associated GM model (upper black: shear; lower black: strain). Spectra are averaged by bins of roughness **(a)**, and by bins of the difference between the beam angle and the bathymetry slope angle **(b)**. For both (a) and (b) number of spectra found by bins and the average bin value are reported in legend. Shear noise spectra associated with a noise velocity of 0.075 m/s are indicated in gray (thickness indicates its association to the N bins). Instrumental wavenumber limits are indicated with the vertical dotted lines: upper limit of $2\pi/128 \; rad \cdot m^{-1}$ for shear due to noise contamination; lower limit of $2\pi/160 \; rad \cdot m^{-1}$ for strain to filter the large-scale background stratification. Retained wavenumber bandwidths for integration are indicated with the shaded ranges on top and bottom (640-107 m for shear; 128-10 m for strain).





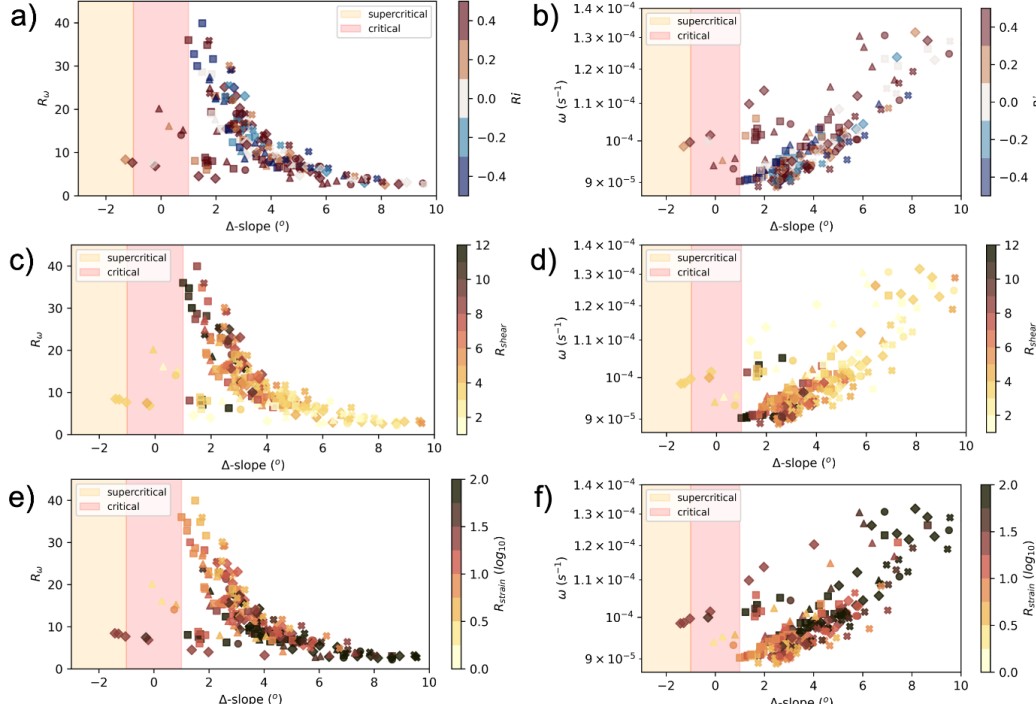

**Figure 5.** Distributions of $R_\omega$ (**left**) and ω (**right**) in the layer 360-920m distant from the bottom, in
function of the difference between the associated beam angle and the bathymetry slope angle. $Z$-axis
is used to indicate **(a,b)** a bulk Richardson number, **(c,d)** the shear-to-shear-GM ratio, **(e,f)** the strain-
to-strain-GM ratio. Higher values on the $x$-axis indicate a beam angle steeper, relative to the
bathymetry slope, while lower values indicate a beam aligning with the slope. The critical range
where the alignment is close to be parallel to the slope is arbitrarily marked by the shaded red area
from -1 to 1 degrees, out of which a supercritical range is indicated in orange.

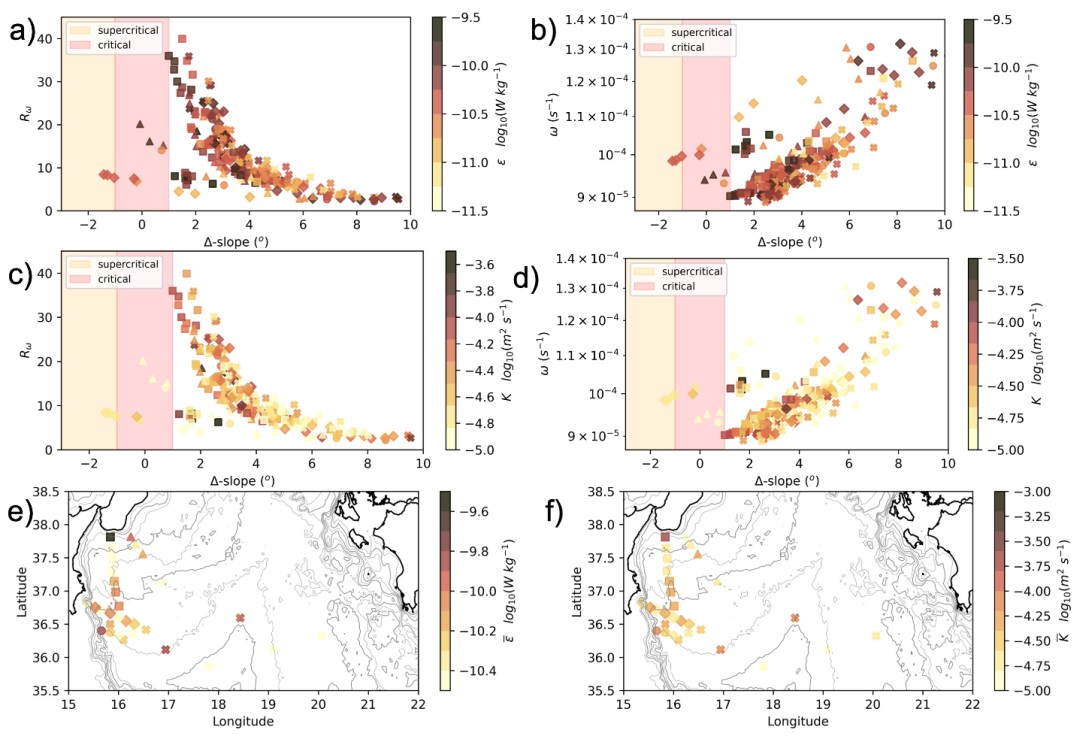

**Figure 6.** Distributions of $R_\omega$ and ω in the layer 360-920 m distant from the bottom, in function of the difference between the associated beam angle and the bathymetry slope angle. Z-axis is used to indicate the turbulent kinetic energy dissipation rate in log10 of $W\ kg^{-1}$ **(a,b)** and diffusion rates in log10 of $m^2\ s^{-1}$ **(c,d)** resulting from the parametrizations. Their respective averages in bins of 80m in the layer 360-920m distant from the bottom are reported over the geographical map of the Ionian Sea (dissipation in **e)**, diffusion in **f)**).



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
