# Peer review of "Bottom topography effects on abyssal diapycnal mixing in the Eastern Mediterranean Sea"

_EGUsphere, 2025_

## Referee Comment (RC1)

Review of MS#egusphere-2025-4762 entitled "Bottom topography effects on abyssal diapycnal mixing in the Eastern Mediterranean Sea" by Florian Kokoszka et al.

This study analyzes the variability of near-bottom shear/strain variance ratios ( $R\omega$ ) in the western Ionian Sea, with particular emphasis on how these ratios are influenced by local seafloor parameters. The authors find that  $R\omega$  is generally low near the shelf break and the Malta Escarpment, in contrast to overall high values observed in the abyssal plain. They demonstrate that lower  $R\omega$  correlates with increased topographic slope and roughness. This is reasonable, given that high-frequency internal lee waves are expected to be generated by the boundary current as it impinges on rough seafloor topography (e.g., St. Laurent et al. 2012; Waterman et al. 2013; Sheen et al. 2013). However, I was unable to find a clear interpretation of these observations in the manuscript. This study would be more strengthened if the authors could include an analysis of lee-wave generation, producing a lee-wave generation map in the Ionian Sea similar to the global map of Nikurashin and Ferrari (2011), and comparing it with the observed  $R\omega$  variability.

In the main part of the analysis, the authors compare  $R\omega$  with the " $\Delta$ -slope", the difference between the  $R\omega$ -derived internal-wave propagation slope ( $\beta$ ) and the local topographic slope, which is conceptually interesting. However, I strongly suspect that the well-defined curves of  $R\omega$  vs  $\Delta$ -slope in Figs 3, 5, 6 arise primarily from the analytic relationship

$$\beta = \tan^{-1}\left(\frac{f}{N}\sqrt{\frac{2}{R_{\omega}-1}}\right),\,$$

with only minimal influence from the actual topographic slope. Moreover, the above analytical relationship of  $\beta$  itself is valid only for internal waves with a "single frequency", and its applicability at low  $R\omega$  has been questioned by Ijichi and Hibiya (2015). Therefore, the discussion in L429-443 regarding whether the topography is near-critical ( $\Delta\text{-slope}\sim0$ ) or super-critical ( $\Delta\text{-slope}<0$ ) is, in my view, questionable, and I do not think it meaningfully aids the interpretation of the observed  $R\omega$  variability. Overall, it is unclear to me what additional insight these  $\Delta\text{-slope}$  plots provide beyond what is already shown in Fig. 2, that is, the relationship between  $R\omega$  and the local topographic slope and roughness.

I also suspect that the authors' method for calculating the local topographic slope and roughness (L393–400) may not adequately distinguish "small-scale roughness" from the underlying "large-scale slope," particularly in regions of abrupt topography such as the shelf break and the Malta Escarpment. One possible improvement would be to estimate the large-scale slope using a least-squares fit, detrend the bathymetry using this large-scale slope, and then compute the roughness from the detrended data.

Lastly, the strain spectra presented in Figs. 1 and 4 become increasingly blue at wavenumbers above ~0.1 cpm. This is unusual for strain spectra derived from standard CTD data (see Kunze et al. 2006) and raises two possibilities: either there are instrument/noise issues in the CTD data used in this study, or the blue spectra reflect real non-internal-wave dynamics (e.g., double-diffusive processes or other high-wavenumber phenomena). The manuscript should address this explicitly. If this is instrumental, the authors must quantify the noise contribution before calculating strain variances. If this is physical, the authors should discuss whether the standard internal-wave finescale parameterizations can be used in this area despite the clear spectral departure.

Because of these major concerns, I am unable to recommend this manuscript for publication in its current form. My additional specific comments and suggestions for future improvement are listed below.

L86: I cannot find the Strait of Gibraltar in Fig. 1.

Fig. 1: It would be helpful to overlay the circulation pattern on the map.

L113-124: are tidal flows much weaker than the mean flow?

L125-134: It would be helpful to explicitly state what is lacking in Artale et al. (2018) in order to better motivate the present study. Clarifying the gaps or limitations in that work will strengthen the rationale for your research.

L206, 208: I suggest reconsidering the use of "overestimate" and "underestimate" in the text. In this context, terms like "increase" and "reduce" may be more accurate and appropriate.

L208-210: I cannot understand this argument. Please explain more.

L249: 0.22, not 0.22N2

L258-261, L403-407: I'm confused with the statements. Could you clarify what "80 m" refers to in this context? Bin size should be larger than 640m?

L268-275: "from 640 to 107 m" It would be easier to understand if you rewrite them in rad/m

L314-373: I think the descriptions are very redundant. For example, equations 7 and 8 are identical. Please rewrite.

L435-438: I think the description is not accurate. The internal-wave frequency or propagating angle doesn't change during the reflection, but wavenumber changes.

L453: is this spectral increase statistically significant?

L480: I cannot discern a clear coherent pattern in the figure. The authors should clarify which aspects they consider coherent and provide an explanation to support this interpretation.

L513: "suppression of the strain in rougher area" is correct?

---

## Author Comment (AC1)

**Reviewer comments**
**Reviewer #1:**

This study analyzes the variability of near-bottom shear/strain variance ratios (Rω) in the western Ionian Sea, with particular emphasis on how these ratios are influenced by local seafloor parameters. The authors find that Rω is generally low near the shelf break and the Malta Escarpment, in contrast to overall high values observed in the abyssal plain. They demonstrate that lower Rω correlates with increased topographic slope and roughness. This is reasonable, given that high-frequency internal lee waves are expected to be generated by the boundary current as it impinges on rough seafloor topography (e.g., St. Laurent et al. 2012; Waterman et al. 2013; Sheen et al. 2013). However, I was unable to find a clear interpretation of these observations in the manuscript. This study would be more strengthened if the authors could include an analysis of lee-wave generation, producing a lee-wave generation map in the Ionian Sea that is similar to the global map of Nikurashin and Ferrari (2011), and comparing it with the observed Rω variability.

*We thank the reviewer for the encouraging comment. We followed the reviewer's suggestion and applied Bell's linear theory to estimate the barotropic-to-internal tide conversion at each station (St. Laurent et al. 2012; See Fig. R1). For this purpose, we used the bottom velocity from the LADCP, the local buoyancy frequency at the seafloor, and an effective topographic height H and length scale L derived from the GEBCO bathymetry (with $H = \sqrt{var(z)}$ and H/L given by the local bottom slope). This yields a Bell-type conversion rate of order $10^{-5}$–$10^{-3}$ $W\,m^{-2}$ over most stations. These small values are consistent with the fact that our $(s/\alpha, ku_0/\omega)$ parameters lie in the subcritical, low-excursion regime, where linear internal tide generation is expected to be weak and lee-wave generation is not dominant. We therefore conclude that barotropic lee-wave generation cannot explain the observed variations of the shear-to-strain ratio, which are more likely related to modifications of the internal-wave spectrum by background stratification and shear.*

[Figure]

***Figure R1.*** *Diagram showing the internal wave response to tidal flow over bathymetry for the slope parameter $s/\alpha$ and the tidal excursion parameter $ku_0/\omega$ (left panel from St. Laurent et al. 2012; middle and right panels from our observations, in the same way than St. Laurent et al. 2012, highlighting variability of topographic roughness and Rω). Internal tides occur when the scale of the topography $k^{-1}$ exceeds the tidal excursion scale $u_0/\omega$. In this regime, waves radiate energy in both the up- and downstream directions, as shown by the group velocity ($c_g$) vectors. Quasi- steady lee waves occur when the scale of tidal excursion exceeds the scale of the topography. Wave energy is radiated in the upstream direction only. The right panel does not show those shallow stations were Rω cannot be estimated.*

In the main part of the analysis, the authors compare Rω with the "Δ-slope", the difference between the Rω-derived internal-wave propagation slope (β) and the local topographic slope, which is conceptually interesting. However, I strongly suspect that the well-defined curves of Rω vs Δ-slope in Figs 3, 5, 6 arise primarily from the analytic relationship

$$\beta = \tan^{-1}\left(\frac{f}{N}\sqrt{\frac{2}{R_\omega - 1}}\right),$$

with only minimal influence from the actual topographic slope. Moreover, the above analytical relationship of β itself is valid only for internal waves with a "single frequency", and its applicability at low Rω has been questioned by Ijichi and Hibiya (2015). Therefore, the discussion in L429-443 regarding whether the topography is near-critical (Δ-slope ~ 0) or super-critical (Δ-slope < 0) is, in my view, questionable, and I do not think it meaningfully aids the interpretation of the observed Rω variability. Overall, it is unclear to me what additional insight these Δ-slope plots provide beyond what is already shown in Fig. 2, that is, the relationship between Rω and the local topographic slope and roughness.

*Many thanks for this comment. Yes, of course, the 1/x behaviour is due to the analytic relationship showed by the reviewer. However, making the difference "Δ-slope" allows us to highlight and disentangle the dynamic range that is controlled by the roughness. This is evident, for instance, in our Fig. 3b (from the original version of the manuscript), where the Rω vs. Δ-slope relation tends to linearize for high values of bathymetric roughness. This confirms that high roughness reduces the Rω range (e.g., Rω in the range 0 - 10) by enhancing higher frequencies, low roughness increases the Rω range (e.g., 0 - 30) by broadening the frequency range, i.e., allowing more low-frequencies. On the other hand, the relation β(Rω) vs. Rω (Fig. R2) does not identify the separation; the 1/x is not what we want to highlight: we want to highlight the dynamical separation.*

[Figure]

***Figure 3b (from the original version of the manuscript)***. *Distributions of Rω in the layer 360-920 m distant from bottom, in function of the difference between the associated beam angle and the bathymetry slope (Δ-slope). Roughness is indicated through the z-axis (in colour). Higher values on the x-axis indicate a beam angle steeper, relative to the bathymetry slope, while lower values indicate a beam aligning with the slope. The critical range where the alignment is close to be parallel to the slope is arbitrarily marked by the shaded red area from -1 to 1 degrees, out of which a supercritical range is indicated in orange.*

[Figure]

***Figure R2.*** *Distributions of Rω in the layer 360-920 m distant from bottom, in function of internal-wave propagation slope (β).*

I also suspect that the authors' method for calculating the local topographic slope and roughness (L393–400) may not adequately distinguish "small-scale roughness" from the underlying "large-scale slope," particularly in regions of abrupt topography such as the shelf break and the Malta Escarpment. One possible improvement would be to estimate the large-scale slope using a least-squares fit, detrend the bathymetry using this large-scale slope, and then compute the roughness from the detrended data.

*We followed the reviewer's suggestion. We computed the bathymetric roughness after removing a large-scale linear trend (least-squares plane fit) and after applying a low-pass Gaussian filter to define the large-scale component (Fig. R3). From a comparison between our original method and what suggested by the reviewer we observe that the station-to-station patterns and the correlations with the shear-to-strain ratio remained essentially unchanged (Figs. R3 and R4). Moreover, the ranking of "rough" versus "smooth" sites was the same as with our original definition. This indicates that our results are robust to the specific detrending/filtering method, and we therefore retain the original roughness metric in the main analysis and mention these tests as a robustness check.*

[Figure]

**Figure R3.** *Setup of the bathymetric smoothing applied to the GEBCO product. The first panel shows the smoothed bathymetry together with a longitudinal section across the Ionian Basin at the mid-latitudes of the study area. The other panels display the slope and roughness computed using spatial windows of 6 points (2.7 km), 12 points (5.3 km), and 23 points (10.3 km). Roughness is estimated using three approaches: (1•) the variance of bathymetry within the window (our current method), (2•) the variance of bathymetry after removing a local smoothing within the window, and (3•) the variance of bathymetry after removing the large-scale smoothing.*

[Figure]

**Figure R4.** *Scatterplots of Rω in function of Δ-slope for the three roughness calculations and for spatial windows of 6, 12, and 23 points.*

Lastly, the strain spectra presented in Figs. 1 and 4 become increasingly blue at wavenumbers above ~0.1 cpm. This is unusual for strain spectra derived from standard CTD data (see Kunze et al. 2006) and raises two possibilities: either there are instrument/noise issues in the CTD data used in this study, or the blue spectra reflect real non-internal-wave dynamics (e.g., double-diffusive processes or other high-wavenumber phenomena). The manuscript should address this explicitly. If this is instrumental, the authors must quantify the noise contribution before calculating strain variances. If this is physical, the authors should discuss whether the standard internal-wave finescale parameterizations can be used in this area despite the clear spectral departure

*To address the reviewer's concern about the blueing of strain at wavenumbers above ~0.1 cpm, we note that 0.1 cpm corresponds to a vertical wavelength of 10 m. This is exactly the upper wavenumber limit we use for finescale integration: **our strain variance is integrated only up to 0.1 cpm**. Therefore, the part of the spectrum where noise might increase is explicitly excluded from the calculation of Rω.*
*The absence of a spectral roll-off simply reflects that we do not use finite differencing (FD) to calculate the strain: instead of employing N², the vertical derivative is obtained by multiplying the density by the vertical wavenumber in spectral space, which avoids the variance loss inherent to FD. This method is a variant from Kunze et al. (2006), and follows Ferron et al. (2014), and it is fully described in our Methods section. Moreover, as described in the methods too, in Ferron et al. (2014) a 160m- high-pass filter is explicitly applied on density to remove large-scale density variations, that can lead to spectral shape differing from Kunze et al. (2006) for scale larger than 160m. The overall level of our strain spectra is also closer to GM, which is consistent with the fact that our observed internal-wave field is close to GM in amplitude. **These considerations will be added in the revised version of the manuscript.***
*We acknowledge that the spectra in Fig. 1 of our original manuscript can be partly misleading because they include all water-column segments, including the shallow ones, which are more likely to contain smaller-scale variance from mesoscale contamination or double-diffusive layers. These segments are not used in our analysis: we focus exclusively on the deepest ~1000 m, and this "non-inclusion" is already visible in the spectra binned by roughness or Δ-slope that encompass only the deepest ~1000 m. **To avoid any ambiguity, we will revise Fig. 1 (see new Fig. R5, which now shows only these analysis-relevant bins).***

[Figure]

***Figure R5.*** *Shear–strain spectra averaged within bins of stratification (upper panel) and computed over the bottom 1000 m of the water column (i.e. excluding surface layers out of the scope of our study). Strain spectra were low-pass filtered at high wavenumbers (cutoff at 10 m) to remove potential noise contamination. Rω distributions are slightly shifted up as a result of the variance filtering, but their overall shape and structure remain essentially unchanged with respect to the original Fig. 1. The associated estimates of Rω are shown by station and by vertical bins of 80 m, within the same bottom 1000 m layer (lower panel).*

*Concerning double-diffusion, we provide a verification showing that its contribution is marginal. The Turner angle (Tu) and the density ratio ($R\rho$) are used to define the local stability of an inviscid water column to double-diffusive*

$$R_\rho = \frac{\alpha\left(\frac{\partial T}{\partial z}\right)}{\beta\left(\frac{\partial S}{\partial z}\right)}; \ Tu = tan^{-1}\left(\frac{R_\rho+1}{R_\rho-1}\right);$$

*where:*

- *-45°<Tu<45°: statically stable*
- *-90°<Tu<-45°: unstable with diffusive convection*
- *45°<Tu<90°: unstable to salt fingering*
- *Tu<-90°, Tu>90°: gravitationally unstable*

*Turner angles (Tu) were computed for all segments, and only a small fraction of the layers is falling into Salt Fingering (SF; 13.6%) or Diffusive Convection (DC; 1.8%) where regimes exhibit active values ( |Tu| above 75º) within the depths where Rω is evaluated. Thus, at the same time than these few potential layers do not present particularly significant structures in density, we consider that double-diffusive layers are only minimally present in our analysis. The corresponding plots are available at the link below:*

*https://drive.google.com/drive/folders/1B_TrGE7PKbZx29nEWeAL285LqKRSoGHk?usp=sharing*